# Chronic Diseases Associated with *Malassezia* Yeast

**DOI:** 10.3390/jof7100855

**Published:** 2021-10-12

**Authors:** Abdourahim Abdillah, Stéphane Ranque

**Affiliations:** 1Aix Marseille Université, Institut de Recherche pour le Développement (IRD), Assistance Publique-Hôpitaux de Marseille (AP-HM), Service de Santé des Armées (SSA), VITROME: Vecteurs—Infections Tropicales et Méditerranéennes, 19-21 Boulevard Jean Moulin, 13005 Marseille, France; abdourahim15@live.fr; 2IHU Méditerranée Infection, 19-21 Boulevard Jean Moulin, 13005 Marseille, France

**Keywords:** *Malassezia*, chronic diseases, associated, body sites, detection

## Abstract

*Malassezia* are a lipid-dependent basidiomycetous yeast of the normal skin microbiome, although *Malassezia* DNA has been recently detected in other body sites and has been associated with certain chronic human diseases. This new perspective raises many questions. Are these yeasts truly present in the investigated body site or were they contaminated by other body sites, adjacent or not? Does this DNA contamination come from living or dead yeast? If these yeasts are alive, do they belong to the resident mycobiota or are they transient colonizers which are not permanently established within these niches? Finally, are these yeasts associated with certain chronic diseases or not? In an attempt to shed light on this knowledge gap, we critically reviewed the 31 published studies focusing on the association of *Malassezia* spp. with chronic human diseases, including psoriasis, atopic dermatitis (AD), chronic rhinosinusitis (CRS), asthma, cystic fibrosis (CF), HIV infection, inflammatory bowel disease (IBD), colorectal cancer (CRC), and neurodegenerative diseases.

## 1. Introduction

*Malassezia* is a genus of fungi which consists of lipid-dependent basidiomycetous yeasts and forms part of the normal cutaneous microbiota of humans and other animals [1,2,3,4]. Although they are a human skin commensal, *Malassezia* species are involved, under certain circumstances, in various skin diseases including pityriasis versicolor, seborrheic dermatitis, folliculitis and dandruff [5]. Infrequently, *Malassezia* yeasts are involved in other dermatological diseases, such as confluent and reticulated papillomatosis and onychomycosis [6]. The association of *Malassezia* with psoriasis and atopic dermatitis has been mentioned and remains a matter of discussion. They have been also involved in fungemia, in immunocompromised and/or neonatal patients, especially those receiving lipidic parenteral nutrition [7,8,9]. In the last decade, the development of culture-independent tools such as high-throughput sequencing has expanded the knowledge of the different niches of *Malassezia*. For example, it is currently known that *Malassezia* spp. distribution is not limited to the skin because DNA-based assays have demonstrated that they are detected at relatively high frequencies in different parts of the body, such as the oral cavity (38% of sequences) [10], the gastrointestinal tract (88% of samples) [11], the respiratory tract (86% of sequences) [12] and the brain (89% of samples) [13]. The detection of *Malassezia* DNA in these different body sites, especially in patients with chronic pathologies, raises many questions. Are these yeasts truly present in the investigated body site or did they contaminate this niche from other body sites? The contamination might be due to the entire yeast (living or dead) or only to DNA. Does this DNA come from living or dead yeast? If these yeasts are alive, do they belong to the resident mycobiota or are they transient colonizers which are not permanently established within these niches? Finally, are these yeasts associated with certain chronic diseases or not? In an attempt to shed light on this knowledge gap, we critically reviewed published studies that analyzed the possible association of *Malassezia* spp. with chronic human diseases.

## 2. Materials and Methods

We conducted literature searches in PubMed (https://pubmed.ncbi.nlm.nih.gov/) and Web of Science (Clarivate Analytics) on 17 May 2021 using the following query: ((*Malassezia* and Diseases) OR (*Malassezia* and Associated) OR (*Malassezia* and Chronic)). No limits were set on publication date or language. The Zotero ver.5 (Corporation for Digital Scholarship; www.zotero.org, accessed on 17 May 2021) software was used to identify duplicates. We screened the articles by analyzing their titles and/or abstracts. 

We only included articles assessing the association or involvement of *Malassezia* in chronic human diseases.

## 3. Results

The search results identified 2538 studies. After removing duplicates, we screened 1790 study titles and/or abstracts and excluded 1728 which were unrelated to the present study. Ultimately, 62 full-length articles were assessed for eligibility, and 31 were selected for qualitative analysis and included in this review. This selection flow is presented in Figure 1. The 31 studies detailed below address the association of *Malassezia* yeast with chronic diseases, including psoriasis, atopic dermatitis (AD), chronic rhinosinusitis (CRS), asthma, cystic fibrosis (CF), HIV infection, inflammatory bowel disease (IBD), colorectal cancer (CRC), and neurodegenerative diseases. We will present the association of *Malassezia* with psoriasis. 

### 3.1. Association between Malassezia and Psoriasis

Psoriasis is a chronic, inflammatory and multifactorial skin disease with an estimated prevalence of 2% [14]. The association of *Malassezia* yeast with psoriasis is widely hypothesized, although the pathophysiological mechanisms are not completely known. The studies reporting the association of *Malassezia* with psoriasis are detailed in Table 1. In some studies, *Malassezia* appears to be increased in terms of the diversity and prevalence of skin culture in psoriatic patients compared to healthy subjects [15,16,17]. Some authors infer a causality from these findings, despite the fact the studies are not always in agreement with one another. This is the case of one study using a nested polymerase chain reaction (PCR) assay in patients in Japan where the average number of *Malassezia* species detected in the skin was found to be high in 22 psoriatic patients (3.7 ± 1.6) compared to 30 healthy individuals (2.8 ± 0.8), suggesting a greater diversity of the skin microbiome in psoriatic patients than in healthy subjects [15]. However, there was no correlation between the species richness found and the severity of the disease, although high detection rates were found in psoriatic patients with 96%, 82% and 64% for *M. restricta*, *M. globosa* and *M. sympodialis*, respectively [15]. A high diversity of *Malassezia* species was also found in 50 psoriatic patients compared to 50 healthy subjects with a positive culture rate of 68% and 60%, respectively [16]. The authors found a higher abundance of colonies isolated from skin lesions (53 colonies) than from normal skin (19 colonies) in psoriatic patients, especially on their scalp (*p* = 0.03), and no difference was found between patients and healthy subjects. In another Spanish study comparing 40 psoriatic patients and 40 healthy subjects, a difference was found regarding the positivity of direct microscopic examinations of scalp samples between patients (75%) and healthy subjects (30%) [17]. These results were confirmed by culture with a positive culture rate of 85% and 50% in patients and healthy subjects, respectively. In addition, the authors found that patients who had suffered an exacerbation of psoriasis had more *Malassezia* than non-exacerbated patients [17]. At the species level, *M. restricta* and *M. globosa* were most frequently isolated, with 40% and 45% in psoriatic patients compared to 25% and 15% in healthy subjects, respectively [17], suggesting an association between the psoriasis and *Malassezia* species. Furthermore, in an ITS1 metabarcoding study of 34 patients with chronic plaque psoriasis compared to 25 healthy subjects, *M. sympodialis* and *M. restricta* were found with a significantly higher abundance and associated with psoriatic lesions on the elbow and the back, respectively [18].

Despite the data from all these studies regarding the distribution of *Malassezia*, the mechanisms by which the *Malassezia* species may initiate or exacerbate psoriasis are not fully elucidated. However, elevated humoral immune responses directed against *Malassezia* have been observed in patients with psoriasis. It has been reported that antibodies against *Malassezia* proteins, such as N-acetyl glucosamine, are present in the serum from patients with psoriasis [19,20]. In addition, T cells which are reactive against *Malassezia* have been isolated from psoriatic skin lesions [21]. It has also been shown that *Malassezia* interacts with keratinocytes and induces an overproduction of molecules involved in cell migration and hyper-proliferation, such as transforming growth factor-beta, integrin chains and heat shock protein 70 [22]. The overproduction of these molecules is likely to promote the exacerbation of the disease in psoriatic patients. Finally, a highly increased *Malassezia*-specific Th1 response was observed in psoriatic patients compared to controls [23]. Interestingly, psoriasis-like lesions were observed when a killed suspension of *Malassezia* was applied to the intact skin of psoriatic patients compared to controls [24], which fulfills one of Koch’s postulates. When taking all these data into account, evidence for the involvement of *Malassezia* in psoriasis flares, particularly in lesions of the scalp, appears to be convincing. However, further evidence is needed because of the discrepancies between the studies regarding the distribution of *Malassezia* species on psoriatic patients’ skin, which are probably due to methodological heterogeneity.

### 3.2. Allergic Diseases Associated with Malassezia

In this section, we will discuss *Malassezia* spp. in allergic diseases, as these yeast species produce allergens and are also found to be associated with some of these diseases, including chronic rhinosinusitis, atopic dermatitis and asthma.

#### 3.2.1. Chronic Rhinosinusitis

Chronic rhinosinusitis (CRS) is a complex inflammatory disease located in the nose and paranasal sinuses with distinctive symptoms such as nasal obstruction, nasal discharge, facial pain and/or a reduced sense of smell [25]. There are three defined phenotypic groups, including CRS with nasal polyps, CRS without nasal polyps, and allergic fungal rhinosinusitis. Although the interactions between the host immune system and microorganisms have not been fully determined, the pathogenic role of the microbiota in CRS has been suggested due to the presence of intramucosal bacteria, biofilms, microbial dysbiosis and super antigens [26]. DNA-based molecular studies have reported a high abundance of *Malassezia* in the sinuses of both CRS patients and controls [12,27,28], generating many questions, as these yeast species were previously only known to colonize the skin. In a 18S ribosomal DNA (rDNA) pyrosequencing study of the sinonasal mycobiome, *Malassezia* was found to be the most abundant fungal genus in 100% of all sinus samples, with a relative abundance of 50.09% and 57.5% in 23 patients with CRS and 11 non-CRS controls, respectively [27]. The presence of *Malassezia* was of great interest as it was the first time it had been described to be associated with the sinuses. The authors hypothesized that it could represent contamination from the nasal vestibule rather than a permanent colonization. Later, Gelber et al. [28] used quantitative polymerase chain reaction (qPCR) to assess whether *Malassezia* was a component of the normal sinus microbiome, to identify *Malassezia* in the sinuses and to compare seven non-CRS controls with CRS subtypes, including 15 CRS patients with polyps, three cases of fungal balls and three cases of allergic fungal rhinosinusitis. In their study, *Malassezia* was detected in 68% of all samples, with an even prevalence across the four groups (*p* > 0.99), suggesting a commensal role for this genus of yeast in the nasal cavity [28]. *M. restricta* was the most commonly detected species with a prevalence of 46% in all subjects, compared to *M. globosa* (14%, *p*= 0.029), which was detected only in CRS patients with polyps [28]. Similar results on the predominance of *Malassezia* were also reported on CRS clinical categories including 50 CRS without nasal polyps, 49 CRS with polyps, seven CRS with cystic fibrosis and 38 control subjects, where *Malassezia* was detected in 100% of subjects with an abundance of 86% of the sequences [12]. These authors found that the predominant *Malassezia* species were distinct in patients with cystic fibrosis compared to the other patients. It is not clear whether particular *Malassezia* species are associated with specific clinical features of CRS. In the case of allergic rhinitis, *Malassezia* was detected by 28 D1/D2 region pyrosequencing on the nasal vestibule with a relative abundance of 93.1% and 97.5% in four patients and four controls, respectively [29]. *M. restricta* was predominant and followed by *M. globosa*, suggesting that these two species are common in the nasal cavity of all clinical categories of CRS. It should be noted that despite the high prevalence of *Malassezia* in these studies, no significant difference was observed between patients and control subjects, reinforcing the idea of the commensal nature of *Malassezia* presence in the upper airways. Specific culture media are not routinely used to isolate *Malassezia* species. It is, therefore, difficult to assess the proportion of viable sinus-associated *Malassezia*. To further investigate this hypothesis, we conducted a study in our laboratory on 14 sinus biopsies from 11 meatus surgery patients. A *Malassezia*-specific real-time PCR was performed [30]. In parallel, the culture method was performed on the sinus samples using the FastFung culture medium that we developed in our laboratory [31,32]. In the 11 samples tested, the PCR was positive in 90.9% of samples, while no sample was positive in culture (presented in Table 2). The negative culture of these samples can be explained by the fact that the samples were inoculated two to three days after surgery, as the post-collection survival time of *Malassezia* in biological samples is not known. Another explanation could be that the DNA detected in the sinuses came from dead yeast.

Little is known about the role and mechanisms in which *Malassezia* might be involved in the sinonasal pathogenesis of patients with CRS. Recently, Lee et al. [33] showed, in a murine model of acute sinusitis, that bacterial-*Malassezia* interactions in the sinonasal mucosa influence the host immune response. The authors showed that co-infection of mice with *M. sympodialis* and *S. aureus* or *p. aeruginosa* increased allergic and inflammatory responses by significantly increasing the level of IL-5, IL-13 and IL-17, respectively [33]. These responses depended on the co-infecting bacterial species and were particularly marked in mice co-infected with *S. aureus*. In addition, the level of expression of the Dectin-1 receptor involved in the fungal sensing was significantly increased when mice were infected with *S. aureus* or *P. aeruginosa* in the presence or absence of *M. sympodialis* compared to mice infected with *M. sympodialis* only [33]. One of the authors’ hypotheses was that the interaction of these bacteria with *Malassezia* present in the sinonasal mucosa could alter the behavior of *Malassezia* by increasing its virulence, thus increasing the expression of Dectin-1 [33]. Since the Dectin-1 receptor recognizes *Malassezia* allergens [34], its activation could be one of the causes for the increase in IL-5 and IL-13 levels via an increased response to Th2-type CD4+ T cell [33]. These data support the hypothesis that *Malassezia* species are natural commensals of the upper airways and that they are probably not capable of exacerbating the disease independently. In other words, the contribution that *Malassezia* makes to the pathogenesis of CRS may be indirect, via interactions with bacteria. Much remains to be known about the impact of *Malassezia* spp. in the sinonasal mucosa of CRS patients. In particular, further assessment is required of the level of expression of *Malassezia* virulence genes when co-cultured with bacteria in the context of the disease.

#### 3.2.2. Atopic Dermatitis and Asthma

Atopic dermatitis (AD) is a chronic inflammatory and relapsing skin disease. It is mainly characterized by intense itching and can be associated with allergic rhinitis and allergic asthma. It has been estimated that in the last three decades, the prevalence of AD has tripled in industrialized countries, affecting 30% of children and 10% of adults [35]. The pathogenesis of AD is complex and not yet fully understood. It has been hypothesized that genetic, immunological and environmental factors and impaired skin barrier function contribute to the disease. Many studies have also highlighted the role of the cutaneous microbiota, including *Malassezia*, in the disease [36]. A few studies have reported high differences in both diversity and abundance of *Malassezia* species colonization in AD patients compared to controls [15,37]. Most studies did not find significant differences between AD and control patients [38,39,40]. This has led some authors to believe that the implication of *Malassezia* in AD may be allergic by nature [38]. Indeed, *Malassezia* displays about 14 known allergens, and each triggers IgE-mediated sensitization in AD patients [5,40]. Many studies, presented in Table 3, which have measured serum-IgE levels, have found significant correlations between the level of *Malassezia*-specific IgE antibodies and the severity of AD [41,42,43,44]. This positive correlation was especially true for AD patients, with localized lesions on the head and neck and very high IgE levels [42,45]. In addition, AD patients treated with ketoconazole experienced a clinical improvement, with a decreased severity score and number of *Malassezia* positive cultures, compared to the placebo group [46]. It was therefore concluded that saprophytic yeasts could be a source of allergens in AD. Currently, there are no solid data showing a direct involvement of allergens in the initiation of AD. However, it is accepted that allergens are factors which exacerbate the disease, as there is an established link between AD severity, IgE levels, and proinflammatory cytokine levels, such as Interleukin (IL)-18 [36].

The pathophysiological mechanisms that *Malassezia* may use to trigger AD are not fully understood. However, two possible mechanisms have been hypothesized to explain *Malassezia*-host interactions. The first involves a direct interaction via recognition of *Malassezia* by pattern-recognizing receptors, such as Mincle and Toll-like receptors [47,48]. The second involves an indirect interaction through recognition of the components released by *Malassezia*. In recent years, many studies have focused on the components released by *Malassezia* and their role in skin inflammation. They have shown that, in addition to the direct release of allergens, *Malassezia* produces extracellular vesicles enriched with allergens, small RNAs and various proteins capable of interacting with skin immune cells [49,50,51]. These extracellular vesicles and allergens have been shown to stimulate keratinocytes and dendritic cells to induce high levels of cytokines including tumor necrosis factor-α (TNF-α), IL-6 and IL-10 [50,52]. Increased levels of these cytokines may be involved in skin inflammation in AD patients. 

AD is also considered a risk factor for the development of atopic respiratory diseases, such as asthma, particularly in children [53]. Since IgE-mediated sensitization to *Malassezia* is found to be high in AD patients, special attention should be given to patients with asthma. Studies on the serology of *Malassezia* IgE in patients with asthma are scarce. However, in one study conducted on 73 AD patients, 156 asthmatic patients and 212 control patients, the prevalence of *Malassezia* IgE was found to be 53%, 1% and 0.5%, respectively [41]. These results did not support a significant association between *Malassezia* and asthma. However, in a 18S rRNA pyrosequencing study of sputum from 30 asthma patients and 13 controls, *Malassezia* was abundantly detected only in asthma patients with a percentage of reads from 0.01% to 21.65% [54]. Another study based on ITS2 metabarcoding recently reported that *Malassezia* was very abundant in sputum specimens from 40 asthmatic patients, especially children, with a mean abundance of 67.69%, 27.04%, 2.02% and 2.94% in pediatric asthma, adult asthma, healthy adult and healthy pediatric participants, respectively [55]. In addition, asthmatic patients receiving a combination of steroid and leukotriene receptor antagonists therapy have a significantly increased abundance of *Malassezia* in their airways. These data are striking in that these molecules are lipid-based while the *Malassezia* species are lipid-dependent. The question of whether the treatment favors the expansion of *Malassezia* in the airways of patients with asthma remains to be clarified. Further studies are needed, including culture and in vivo and in vitro experiments, to understand the pathophysiology of *Malassezia* in asthma patients.

### 3.3. Cystic Fibrosis and Malassezia

Similarly to the gastrointestinal tract, the respiratory tract is a complex polymicrobial niche which includes various bacteria, fungi and viruses [56]. These microorganisms can exacerbate chronic pulmonary diseases including cystic fibrosis (CF). A growing number of DNA-based studies reported high detection frequencies of *Malassezia* in the respiratory tract of patients with CF. The first molecular study of *Malassezia* in CF patients resulted from a study comparing culture and DNA-based methods for fungal detection in 77 adult patients with CF [57]. *Malassezia* was detected in 2/77 patients with CF by sequencing targeting the ITS2 region. An increasing number of subsequent study reported a relatively high prevalence rates of *Malassezia*. In a study of four French CF patients, Delhaes et al. [58] studied the fungal community of eight sputum samples. They found a high number of *Malassezia* reads (≈0.015% of fungal reads) in all patients by ITS2 pyrosequencing. At the species level, *M. restricta* was detected in all patients while *M. globosa* and *M. sympodialis* were detected in only one patient. In their study, they also used mycological culture media, but none were appropriate for *Malassezia* spp. which require specific media. Similar results were found on four CF patients by analyzing the ITS1 region by denaturing high-performance liquid chromatography combined with cloning/sequencing, where *M. restricta* and *M. globosa* were the two species identified on the sputum samples from three patients [59]. Another study carried out on six CF patients receiving antibacterial treatment also found an enrichment of a mixture of *Candida* species and *Malassezia* in all sputum samples, representing 74–99% fungal reads using a ITS1 pyrosequencing-based analysis [60]. These authors found that the abundance of both fungal genera persisted before and after treatment suggesting stability.

Although the sample sizes and participants in these studies were small, the results converge, suggesting that these common identified fungal species play a potential role in the respiratory tract of patients with CF. For *C. albicans*, it has already been shown by culture that this yeast species colonizes the airways of between 49.4% and 74.3% of patients in large cohorts of CF patients [61,62]. In addition, this colonization is associated with the exacerbation of the disease and impaired lung function [61], although the pathogenic mechanisms are not yet elucidated. For *Malassezia* spp., many questions remain unanswered about the *Malassezia* species detected in respiratory samples. It is unclear whether the DNA detected comes from dead cells or whether *Malassezia* survives as a commensal in the lungs. In the literature, isolation of *Malassezia* spp. from the respiratory tract has been reported in immunocompromised patients and was associated with clinical deterioration [63,64], suggesting that the high detection of *Malassezia* spp. in CF patients is deleterious and deserves further investigation. Recently, an ITS2 metagenomic sequencing study analyzed 33 sputum samples from patients with and without CF pulmonary exacerbation [65]. Interestingly, *Malassezia* was abundant and associated with CF pulmonary exacerbation. Further studies using in vivo and in vitro models will be necessary to understand the potential pathophysiological mechanisms.

### 3.4. Malassezia and HIV Infection

Patients living with HIV are subject to several microbial infections secondary to their immunosuppression. These infections can be caused by commensal opportunistic pathogens in different anatomical sites, such as the skin, a localization where *Malassezia* is the dominant genus of the human skin mycobiota [3]. With incidence of up to 80%, seborrheic dermatitis (SD) due to *Malassezia* is one of the major diseases affecting HIV patients [66,67]. In some quantitative culture studies, the density of *Malassezia* was found to be higher in the skin of HIV patients compared to non-HIV controls [68]. One recent study in particular reported that the positive culture rate and density of *Malassezia* were very high in 30 HIV patients with seborrheic dermatitis (76.7%) compared to 30 HIV patients without seborrheic dermatitis (50%) [69]. This high rate could be explained by the number of CD4 cells present in the patients, as in the seborrheic dermatitis patients, 24 had <500 CD4 cells including nine with <200 CD4 cells. Moreover, it has been reported that the CD4 count is a primary factor associated with the severity of seborrheic dermatitis in HIV patients, as it has been found to be associated with the abundance of yeast in skin lesions, the severity of seborrheic dermatitis and the CD4-positive T lymphocyte count [70]. This shows that CD4 cell-mediated immunity plays a crucial role in controlling the switch of these yeasts from commensal to pathogen. Whether other non-immune factors related to HIV promote the overgrowth of *Malassezia* remains to be determined. Regarding the difference in the prevalence of *Malassezia* between HIV and non-HIV patients, a recent study reported a positive culture rate of 69% and 79% in 48 HIV and 48 non-HIV patients, respectively [71]. They found that patients with high CD4 cells had relatively high colonization rates, but the difference with the other groups was not statistically significant. To date, there are no data showing any other consequence of overgrowth of *Malassezia* spp. in HIV patients outside the skin. However, by studying the gut eukaryotic microbiota using ITS metabarcoding in our laboratory, we found that HIV patients have a higher abundance of *Malassezia* spp. in their gut compared to healthy subjects [72]. It is not clear whether this high abundance is a common phenomenon in all patients living with HIV or whether it is a coincidence. Remarkably, in the 22/31 HIV patients in whom *Malassezia* spp. was detected, 15 patients had CD4 counts ranging from 2–436 cells, with an average of 149.73 cells. This suggests that the outcome of HIV-associated immune deficiency on *Malassezia* spp. is not limited to the skin and that *Malassezia* spp. probably colonizes other internal organs in immunocompromised HIV patients. Future studies using complementary techniques, including culture, are needed to further investigate the potential impact of *Malassezia* spp. on HIV patients, particularly within the gut and internal organs.

### 3.5. Malassezia in Enteric Diseases

*Malassezia* is increasingly detected at a relatively high prevalence in the gastrointestinal tract both in healthy individuals at a rate of 81–88.3% [11,73] and in diseased individuals at a rate of 68.4–100% [72,74,75]. In this section, we will discuss the potential role of *Malassezia* in the gastrointestinal tract in several enteric diseases, including inflammatory bowel disease and colorectal cancer.

#### 3.5.1. Inflammatory Bowel Disease

Inflammatory bowel disease, which includes Crohn’s disease and ulcerative colitis, is a chronic inflammatory disease of the gastrointestinal tract. The gut microbiota and environmental and genetic factors influence inflammatory bowel disease although their exact role has not yet been completely determined. Little is known about the role of the gut mycobiota compared to bacterial gut microbiota in these diseases. The role of fungi has begun to generate great interest in research since high levels of anti-*Saccharomyces cerevisiae* antibodies (ASCA) have been found in the sera of patients with Crohn’s disease [76]. In addition, genome-wide association studies have identified variants in the genes coding for molecules involved in the defense against fungal infections, such as the caspase recruitment domain 9 (CARD9) [77]. By using sequencing methods, alterations of the gut mycobiota in patients with inflammatory bowel disease were observed with a high relative prevalence of yeasts such as *Candida*, *Saccharomyces* and *Malassezia* genera [74,78]. The role of these yeasts in inflammatory bowel disease pathogenesis is not fully understood. However, two studies have shown that *C. albicans* and *S. cerevisiae* exacerbate inflammation and colitis in mouse models with chemically induced colitis, suggesting that these members of the gut mycobiota may be associated with the severity of inflammatory bowel disease [79,80]. 

It has long been thought that *Malassezia* species were limited to the skin microbiota. For this reason, there was no interest in investigating whether they were involved in other diseases besides the skin, particularly in internal organs. Recently, Limon et al. [81] sought to understand the role of *Malassezia* in Crohn’s disease after the presence of *M. restricta* in the gut mucosa had been found to be associated with Crohn’s disease. In their study, they found that the presence of *Malassezia* was strongly associated with the Crohn’s disease risk allele CARD9S12N, and this association was correlated with an increased abundance of *Malassezia* in sigmoid colon samples of Crohn’s disease patients. In addition, depending on CARD9 signaling, they found that *M. restricta* triggers a stronger inflammatory response from innate immune cells (human or mouse dendritic cells) harboring the IBD-associated polymorphism, compared to *C. albicans* and *S. cerevisiae*. By giving pathogen-free mice an oral gavage with *M. restricta*, the authors found that this yeast species causes a shortening of the colon, a worsening of disease activity, and more severe intestinal inflammation characterized by increased mucosal erosion, crypt destruction, and inflammatory cell infiltration. In addition, in pathogen-free mice colonized with eight specific bacteria, *M. restricta* exacerbated colitis without altering the bacterial microbiota, showing that *M. restricta* is sufficient to exacerbate the disease by activating the intestinal immune response. One of the most striking results was the correlation of ASCA levels with *M. restricta*-recognizing IgA and IgG from the sera of patients with Crohn’s disease. The findings of this study showed that in patients with Crohn’s disease, *M. restricta* may increase disease severity, principally in patients carrying the CARD9S12N allele.

The high detection of *Malassezia* in the digestive tract has always raised questions, as this yeast is known to belong to the skin mycobiota. The question of whether it is a true living colonizer of the digestive tract remains a subject of debate. Although data from mouse models are difficult to extrapolate to humans, the presence of *Malassezia* had significant consequences on the outcomes of Crohn’s disease models. Therefore, particular attention should be paid to fungal gut alterations in patients with IBD.

#### 3.5.2. Colorectal Cancer

Colorectal cancer is one of the most frequent chronic diseases affecting the digestive tract, and the gut microbiota can contribute towards this disease [82]. As with the bacterial microbiota, the existing studies on the gut mycobiota show fungal dysbiosis in colorectal cancer patients with an enrichment of certain fungal genera including *Malassezia*. In a Chinese study conducted by Gao et al. [83], the gut mycobiota was analyzed by sequencing stool samples from 74 colorectal cancer patients, 29 colon-polyp patients and 28 healthy controls. The impacts of anatomic position and tumor stage on the fungal dysbiosis were also investigated. The authors found fungal dysbiosis in colon polyps and colorectal cancer with decreased diversity in polyps patients. An increased proportion of opportunistic fungi, including *Trichosporon* and *Malassezia* were observed in stool samples of patients compared to controls. They concluded that these opportunistic fungi and altered biodiversity may favor the progression of CRC. Similar results were found by analyzing the shotgun metagenomic sequences in fecal samples of 73 patients with colorectal cancer and 92 control subjects [84]. The authors identified fungal biomarkers associated with colorectal cancer including *Malassezia* with a higher prevalence in colorectal cancer patients compared to controls. They then further validated their results in an independent cohort of 90 patients with CRC, 42 patients with adenoma, and 66 control subjects of published repository sequences from Germany and France. This observed association supports the idea that *Malassezia* may be a biomarker and contribute to colorectal cancer tumorigenesis.

The mechanisms by which *Malassezia* may contribute to the tumorigenesis of colorectal cancer remain unknown. We found no studies on this topic. However, the contribution of *Malassezia* in tumorigenesis has been shown by Aykut et al. [85] in mice models with pancreatic ductal adenocarcinoma. In their study, the authors found that the tumor mycobiome was distinct from that of the gut or normal pancreas and that *Malassezia* spp. were very abundant in pancreatic tumor tissues. Using mouse models, they showed that fungi, including *Malassezia* spp., can migrate from the gut to the pancreas and can accelerate the progression of ductal pancreatic adenocarcinoma via the complement cascade activation. Interestingly, pancreatic oncogenesis was strongly increased by *Malassezia* compared to other common fungi, such as *Candida*, *Saccharomyces* and *Aspergillus*. Currently, there are no established hypotheses on the potential mechanisms of *Malassezia* involvement in cancer, and further studies are needed to investigate the association of *Malassezia* with CRC.

### 3.6. Malassezia Neuroinfection in Neurodegenerative Diseases

Neurodegenerative diseases, including Alzheimer’s disease, amyotrophic lateral sclerosis, Parkinson’s disease, and multiple sclerosis, are multifactorial and represent a major human health issue. Although these diseases share some common features, as the deposition of misfolded protein aggregates in distinct central nervous system (CNS) regions, the exact etiological causes are partially known and remain a subject of research. The idea that infectious agents may be among the etiological causes or risk factors for neurodegeneration has been widely discussed and numerous studies have been conducted on viral and bacterial infections [86,87,88]. The neuroinfection by fungi in neurodegenerative diseases has triggered a considerable interest in research since the dawn of next-generation sequencing (NGS) technology. In recent years, a Spanish team has conducted intense research aiming to highlight the possible involvement of fungi in neurodegenerative diseases [13,89,90]. In their studies, they found common fungi that are more abundant in the affected areas of the brain in patients compared to healthy subjects. Among these fungi, *Malassezia* was widely detected. In the case of Alzheimer’s disease for example, *Malassezia* was detected by NGS on brain tissue samples in 94.7% of patients, compared to 12.5% of the controls, in two studies involving 19 patients and 16 healthy controls [89,90]. By comparing the results of Alzheimer’s disease patients with younger and elderly controls, they found that the percentage of *Malassezia* was very high in the Alzheimer’s disease patients (≈4.4%) compared to the younger and elderly controls ( < 0.5%) [90]. In younger and elderly controls, this low percentage of *Malassezia* suggests that the high burden of *Malassezia* in the CNS of patients is probably related to the disease and not to age. The authors argued that these results could be used to guide targeted antifungal therapy for Alzheimer’s disease patients. This will serve to answer one of the most important questions, namely, are disseminated fungal infections the cause of Alzheimer’s disease, a risk factor, or simply a consequence of neurodegeneration? [89]. In addition to Alzheimer’s disease, a high *Malassezia* detection rate was also found in the CNS of patients with amyotrophic lateral sclerosis (100% of 11 patients), Parkinson’s disease (100% of six patients) and multiple sclerosis (90% of 10 patients) patients [13,91,92]. All these studies aimed to demonstrate the presence of fungi in the altered areas of the CNS. However, based on these data, we cannot conclude whether fungi, including *Malassezia*, have an etiological responsibility or play a role in the exacerbation of neurodegenerative diseases. All that can be currently concluded is that there is a relatively high *Malassezia* colonization rate in the CNS of patients who suffer from neurodegenerative diseases. 

To date, our knowledge on the possible pathogenesis mechanisms involving fungi such as *Malassezia* in CNS patients with neurodegenerative diseases is limited. It remains to be clarified whether fungal infections of the CNS are the consequence of immune dysfunctions and/or the patients’ genetic background. It was shown that CARD9 plays a critical role in the control of fungal invasion by recruiting neutrophils into the CNS, knowing that CARD9-deficient humans develop fungal infections targeting the CNS [93]. CARD9 is one of the crucial receptors involved in antifungal immunity as shown in other diseases such as Crohn’s disease [81]. In CNS infection, microglial cells actively participate in the production of proinflammatory cytokines. It was shown that microglial IL-1β and CXCL1 production depends on CARD9, and the specific deletion of microglial CARD9 impairs the neutrophil recruitment toward the *C. albicans*–infected CNS [94]. Given these data, it is plausible that the fungal infections observed in patients with neurodegenerative diseases could be due to deficiencies in the molecules involved in antifungal immunity.

## 4. Discussion

It is recognized that the skin is the main ecological niche of *Malassezia*, and its responsibility for some skin infections, such as pityriasis versicolor and seborrheic dermatitis, is no longer the subject of debate. The different anatomical sites, outside the skin, in which *Malassezia* has been detected, either in the context of chronic diseases or in healthy individuals, are illustrated in Figure 2. However, its causative role in other inflammatory skin diseases including psoriasis and atopic dermatitis remain under discussion. Given the multifactorial cause of these two diseases, the exact role of *Malassezia* in their physiopathogenesis remains to be clarified. Regarding psoriasis, *Malassezia* overgrowth, especially within skin lesions (Table 1) suggests that *Malassezia* may play a role in its exacerbation [16,17]. However, whether *Malassezia* plays pathogenic role by causing the lesions or whether it is a simple opportunistic colonizer of pre-existing psoriasis lesions remain under discussion. Regarding atopic dermatitis, we found no evidence of any link between *Malassezia* abundance on the skin and the severity of this disease. In contrast, increased anti-*Malassezia* IgE antibodies levels have been associated with AD flare-ups (Table 3).

Independently of seborrheic dermatitis in HIV patients, there are conflicting results regarding *Malassezia* skin colonization, which are probably due to the heterogeneity in the diagnostic procedures used or in the patients’ profiles, including their compliance with antiviral treatments. It is difficult to evaluate the impact of *Malassezia* skin colonization on HIV infection. However, the abundance of *Malassezia* within the gastrointestinal tract deserves a special attention, since most patients had relatively low CD4 cell counts [72]. HIV infection is associated with a depletion of CD4+ cells, especially in Th17 within the gastrointestinal tract [71,95]. Because Th17 cells are known to participate in the immunological response against *Malassezia* in the skin [71], this intestinal immune dysregulation in HIV patients may be one explanation for the overgrowth of *Malassezia* in the gastrointestinal tract. This hypothesis remains to be confirmed in future studies which should focus on both the clinical and immunological impacts of the presence of *Malassezia* in the gastrointestinal tract of HIV patients. Regarding Crohn’s disease, *Malassezia* has been shown to play a role in inflammation using human samples and a mouse model of induced colitis [81]. This role has been explained by the fact that *Malassezia* produces ligands at the aryl hydrocarbon receptor, which is involved in several functions such as skin homeostasis or the production of proinflammatory cytokines [81]. Moreover, *Malassezia* was found to be remarkably abundant within the digestive tract of colorectal cancer patients, suggesting that it might be used as a colorectal cancer biomarker [83,84]. Although there are no studies on the link between *Malassezia* and the pathogenesis of CRC, it has been shown that *Malassezia* can migrate from the gut to other organs, such as the pancreas, where it is involved in pancreatic tumorigenesis [85].

Additionally, DNA-based analysis of the upper and lower respiratory mycobiota has also shown that *Malassezia* is highly prevalent in both healthy and diseased subjects. In particular, in the case of CF, the abundance of *Malassezia* has been associated with CF pulmonary exacerbation [65]. There are no studies on the possible pathophysiological mechanisms of *Malassezia* on the lungs. Most studies found no difference in the prevalence of *Malassezia* between CRS patients and healthy subjects in the sinuses, which was consistent with its commensal nature [33].

Regarding neurovegetative diseases, *Malassezia* was found to be highly abundant in CNS patients [89,90]. It should be borne in mind that all the arguments and evidence provided in these studies have sought to highlight the existence of fungal infections and have not targeted any specific fungal agent. It is, therefore, difficult to identify a significant contribution of each fungal species detected in neuropathology to these different neurodegenerative diseases. Furthermore, these results on fungal colonization in CNS patients remain to be confirmed by other independent teams around the world. If confirmed, one particular consideration would appear to be essential to understand the role and contribution of commonly found fungi such as *Malassezia* spp. If truly present, is the fungus the inciting force for disease, is it incidental, or does it function in an additive/synergistic manner that leads to disease progression? Are *Malassezia* the instigator of the disease or an opportunist that colonizes tissues that have been injured by another factor? In the future, it would be interesting, for example, to measure antibodies against *Malassezia* in the serum of patients, to use a specific probe for fluorescence in situ hybridization (FISH), as well as to use specific anti-*Malassezia* immunostaining or auto-immunohistochemistry to thoroughly investigate *Malassezia* spp. in these diseases. Animal models of neurodegenerative diseases infected by *Malassezia* spp. may provide greater insight into neurological infections. The study of the genetic background of patients may also be important in terms of better understanding these infections.

Overall, our literature review highlighted some important information. First, although the studies tend to transpose the results of the mice models to humans, it is necessary to consider this with hindsight, because human and animal microbiota are different. Furthermore, it is easy to inoculate a known amount of a microorganism into a mouse model and to measure the effects on the host. However, in the clinical setting, it is difficult to assess the number of microorganisms present in a given site at a given time. Second, it should be noted that there are methodological biases from one study to another, which are due to the heterogeneity of either the sampling methods, DNA extraction, DNA-based detection methods, or culture methods used. With respect to DNA-based approaches, some studies have targeted the 18S rRNA gene, while others targeted either the ITS1 or the ITS2 regions and rarely both at the same time. However, there may be an amplification bias of either of two regions on detected fungi [96,97]. In addition, most of these DNA-based assays have either detected or not detected all the DNA that was present in the samples, and they do not distinguish DNA from dead or live cells. This can be problematic because the detection of a microorganism’s DNA does not necessarily link this microorganism with the disease. In the clinical setting, the detected DNA may originate from dead cells or from contamination. The problem of contamination is likely to be excluded because the studies were conducted by different teams around the world. The issue of the viability of *Malassezia* detected on these different niches must be elucidated. Since *Malassezia*-specific culture media are not routinely used, especially for internal diseases, the use of molecular viability techniques, such as viability PCR with propidium monoazide, could be used as a solution.

## 5. Conclusions

The presence of *Malassezia* yeast, with various levels of abundance, appears to be ubiquitous in the human body and may be associated with several diseases of the internal organs including Crohn’s disease, CRS, diseases of the respiratory tract and, probably, neurovegetative diseases. It has been shown to be involved in psoriasis and atopic dermatitis flare-ups, which is not surprising, as these yeasts are a resident member of the skin mycobiome. Independently of SD, *Malassezia* colonization appears to be associated with the CD4 count in patients living with HIV. Whether *Malassezia* is an emerging pathogen or an ancient under-diagnosed pathogen involved in diseases of the internal organs remains to be elucidated, and further studies are needed to fill the knowledge gaps on the pathophysiology of the presence of *Malassezia* yeast in the different human organs.

## Figures and Tables

**Figure 1 jof-07-00855-f001:**
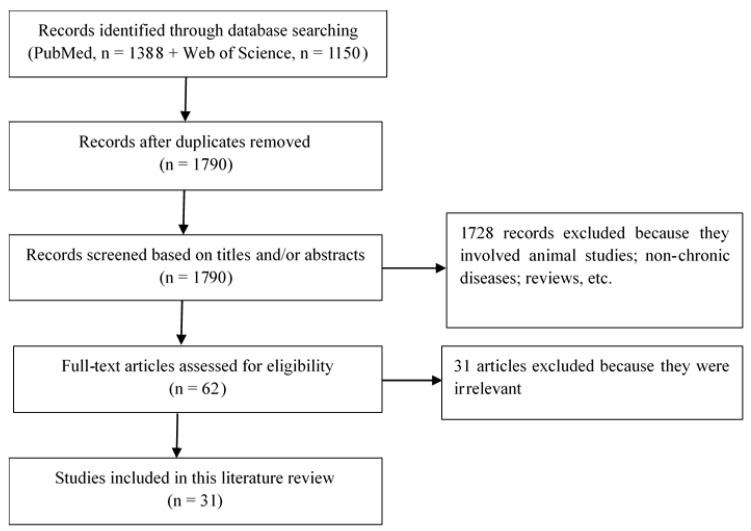
Selection flow-chart of the studies included in the review.

**Figure 2 jof-07-00855-f002:**
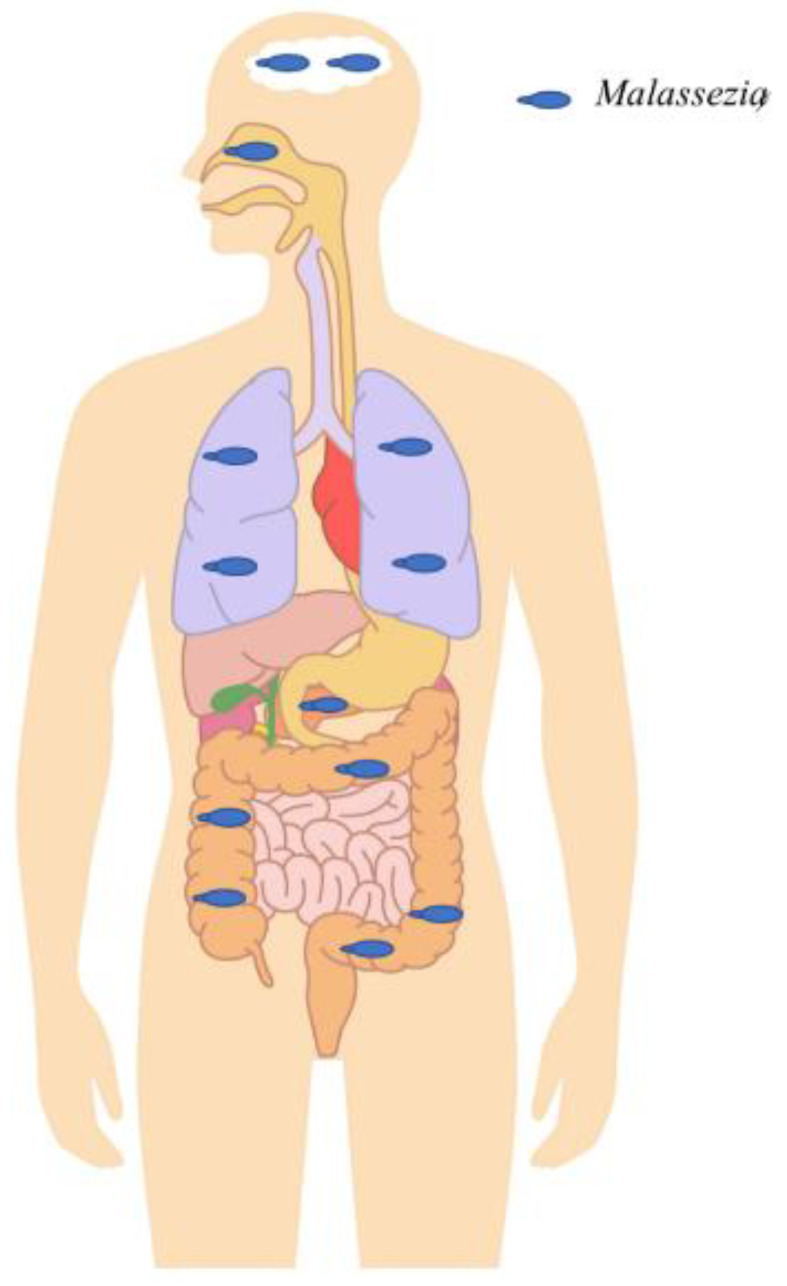
*Malassezia* in different anatomical sites detected in the context of chronic diseases or healthy individuals outside the skin. *Malassezia* was abundantly detected in the digestive tract and was shown to be associated with Crohn’s disease by exacerbating colitis and inflammation. Furthermore, it was highly abundant in colorectal cancer patients compared to healthy subjects, suggesting a potential role in tumorigenesis. It was also shown to be able to migrate from the gastrointestinal tract lumen to the pancreas and to exacerbate pancreatic adenocarcinoma progression. High abundance was also found in the gastrointestinal tracts of HIV patients, probably due to HIV-related immunosuppression. It was highly detected in the upper and lower respiratory tract. Within the upper respiratory tract, it has been shown to interact with specific bacteria (*Pseudomonas aeruginosa* and *Staphylococcus aureus*) and to increase chronic rhinosinusitis-associated inflammation. Within the lower respiratory tract, it was shown to be associated with cystic fibrosis pulmonary exacerbations. Finally, *Malassezia* were detected with a relatively high prevalence and abundance in brain tissue samples of patients with neurodegenerative diseases, such as Alzheimer’s disease, Parkinson’s disease, etc.

**Table 1 jof-07-00855-t001:** Studies showing the association of *Malassezia* with psoriasis.

Psoriasis Patients (PP) and/or Controls	Samples	Methods	*Malassezia*	References
22 PP vs. 30 healthy subjects (HS)	lesional and non-lesional skin	Nested PCR	High detection (64–96%) in PP Species diversity (3.7 ± 1.6 in PP vs. 2.8 ± 0.8 in HS)	[15]
50 PP vs. 50 HS	lesional and non-lesional skin	Culture/ITS2 PCR-RFLP and D1/D2 sequencing	68% of PP cases vs. 60% of HS In PP, more colonies isolated in psoriatic lesions (53 colonies) than in non-lesional skin (19), especially on the scalp (*p* = 0.03). High species richness in PP than HS	[16]
34 PP vs. 25 HS	skin lesion and non-lesion	ITS1 metabarcoding	A significantly higher abundance of *M. restricta* and *M. sympodialis* in psoriatic lesions than HS	[18]

**Table 2 jof-07-00855-t002:** Culture and PCR results on sinus biopsies.

Patients	Sex	Age (Years)	Samples	Culture	*Malassezia* PCR
Patient 1	Female	77	Sinus	Negative	Positive
Patient 2	Female	27	Sinus	Negative	Positive
Patient 3	Male	52	Sinus	Negative	Positive
Patient 4	Male	56	Sinus	Negative	Positive
Patient 5	Female	44	Sinus	Negative	Positive
Patient 6	Female	53	Sinus	Negative	Positive
Patient 7	Female	13	Sinus 1	Negative	Positive
			Sinus 2	Negative	Positive
Patient 8	Male	48	Sinus	Negative	Negative
Patient 9	Male	25	Sinus 1	Negative	Positive
			Sinus 2	Negative	Positive
Patient 10	Female	53	Sinus	Negative	NA
Patient 11	Female	72	Sinus 1	Negative	NA
			Sinus 2	Negative	NA

NA: not applicable; DNA not available.

**Table 3 jof-07-00855-t003:** Studies associating *Malassezia* with allergic diseases.

Patients and/or Controls	Samples	Methods	*Malassezia*	References
23 CRS vs. 11 controls	Sinonasal swabs	18S pyrosequencing	100% of all sinus samples Relative abundance of 50.09% in CRS patients vs. 57.5% in 11 controls	[27]
21 CRS vs. seven controls	Sinus brushings	qPCR	68% of all samples with no prevalence variation among the groups (*p* > 0.99) *M. restricta* (46%) than *M. globosa* (14%, *p* = 0.029)	[28]
106 CRS vs. 38 controls	Mucosal swabs	ITS2 metabarcoding	100% of subjects with an abundance of 86% of the sequences	[12]
56 AD vs. 32 controls	lesional and non-lesional skin	qPCR	High *Malassezia* colonization in patients with severe AD, ≈ two- to fivefold that in mild and moderate AD patients and healthy subjects (*p* < 0.05)	[37]
106 head and neck AD (HNAD) vs. 61 controls	Blood	Anti-*Malassezia* IgE assays (Pharmacia CAP System)	Significant correlation between *M.* *furfur* IgE levels and severity in HNAD patients (*p* < 0.0001)	[42]
63 AD vs. 23 controls	Blood	Anti-*M. globosa* IgE by Enzyme-linked immunosorbent assay (ELISA)	High significantly IgE and correlated with severity in AD patients compared to normal controls (*p* < 0.001)	[43]
53 severe AD and 126 AD moderate vs. 140 controls	Blood	Anti-*M. sympodialis* IgE (ImmunoCAP™)	62% of severe AD compared to 39% of moderate AD (*p* < 0.01), No positive controls	[44]
74 HNAD vs. 99 Non-HNAD	Blood	Anti-*Malassezia* IgE (ImmunoCAP™)	Significantly higher levels of IgE in HNAD patients than non-HAND (*p* < 0.001)	[45]
73 AD, 156 asthmatic and 212 control patients	Blood	Anti-*M. furfur* IgE Fluoroimmuno assay (CAP system)	AD (53%), asthmatic (1%) and non-asthmatic control subjects (0.5%)	[41]
30 asthma patients vs. 13 controls	Sputum	18S pyrosequencing	Only in asthma patients with a percentage of reads from 0.012% to 21.651%	[54]
21 asthma patients vs. 19 controls	Sputum	ITS2 metabarcoding	Mean abundance of 67.69%, 27.04%, 2.02% and 2.94% in pediatric asthma, adult asthma, healthy adult, and healthy pediatric participants, respectively. Significant abundance of *Malassezia* in the airways of asthmatic patients receiving steroid therapy combined with leukotriene receptor antagonists.	[55]

## Data Availability

Not applicable.

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
