# Peer review of "Chronic Diseases Associated with Malassezia Yeast"

_jof, 2021, doi:10.3390/jof7100855_

Round 1

Reviewer 1 Report

The submitted review article summarizes the current knowledge focused on the association of Malassezia spp. with chronic human diseases such as psoriasis, atopic dermatitis, chronic rhinosinusitis, asthma, cystic fibrosis, HIV infection, inflammatory bowel disease, colorectal cancer, and neurodegenerative diseases. The chosen topic is interesting for experts who deal with similar issues. The results include data analysed on the base of 31 selected literature sources. Each subchapter provides a relevant explanation of the role of Malassezia yeasts in selected chronic diseases.

The attached figures are useful complement for understanding the described results.

The review article is well written, clear and easy to read.

The conclusions point to the most important findings of the study. They are in line with the arguments describe in this article.

I have no serious comments to the reviewed article.

Minor revisions listed in the comment for authors are required.

Comments for authors:

Please, make the following corrections to this article:

Line 66: Change the number in Figure 1 to number 2.

Line 90: …a exacerbation…, correct: …an exacerbation…

Check the hyphenation at the end of lines 129, 131,160,216, 235, 241, 253,394, 406, 458, 480, 506, 513, 525, 531, 533, 571.

Author Response

Please, make the following corrections to this article:

  • Line 66: Change the number in Figure 1 to number 2.
  • R: We have delete the reference to Figure 2 in the introduction section and changed the figure number.
  • Line 90: …a exacerbation…, correct: …an exacerbation…
  • R: Corrected
  • Check the hyphenation at the end of lines 129, 131,160,216, 235, 241, 253,394, 406, 458, 480, 506, 513, 525, 531, 533, 571.
  • R: We saw no issue with the hyphenation; but the editor might further check.

Reviewer 2 Report

REPORT FOR TRANSMISSION TO AUTHORS

In this article, the authors provide a comprehensive overview of current studies on the possible association of Malassezia spp. – a lipid-dependent basidiomycetous yeast found on all skin surfaces of humans- with chronic human diseases. including psoriasis, atopic dermatitis, chronic rhinosinusitis, asthma, cystic fibrosis, HIV infection, inflammatory bowel disease, colorectal cancer and and neurodegenerative diseases. In addition, the focus of this article is to shed light on the pathophysiology of the presence of Malassezia yeast in the different human organs. The manuscript is is nicely presented, well written and easy to understand. Before publication should be considered, however, I have a few suggestions on how the manuscript could be improved prior to publication.

  • The authors need to discuss in more detail why Malssezzia spp and not other fungi with similar characteristics should play an important role in the onset/evolution of chronic human diseases. If there is no scientific evidence the authors should speculate based on the unique characteristics of this species
  • The authors should specify in more detail the role played or that could play this fungus in each of the chronic diseases mentioned

Author Response

  • Reviewer 2

    In this article, the authors provide a comprehensive overview of current studies on the possible association of Malassezia spp. – a lipid-dependent basidiomycetous yeast found on all skin surfaces of humans- with chronic human diseases. including psoriasis, atopic dermatitis, chronic rhinosinusitis, asthma, cystic fibrosis, HIV infection, inflammatory bowel disease, colorectal cancer and and neurodegenerative diseases. In addition, the focus of this article is to shed light on the pathophysiology of the presence of Malassezia yeast in the different human organs. The manuscript is is nicely presented, well written and easy to understand. Before publication should be considered, however, I have a few suggestions on how the manuscript could be improved prior to publication.

    • The authors need to discuss in more detail why Malssezzia spp and not other fungi with similar characteristics should play an important role in the onset/evolution of chronic human diseases. If there is no scientific evidence the authors should speculate based on the unique characteristics of this species
    • R: The way we query the literature is unquestionably biased towards finding Malassezia associated diseases. It would be different if we had used the keyword “fungi” instead of “Malassezia”, but it would be another review.
    • The authors should specify in more detail the role played or that could play this fungus in each of the chronic diseases mentioned
    • R: We agree that the imputatbility to Malassezia of any diseases that we reviewed in our manuscript is a critical point. We have developed in the text all available evidences.

Reviewer 3 Report

This is an interesting review of the links between Malassezia and a variety of chronic diseases. Some of the associations are better linked than others, and this is a rich area for further study.

The review contributes to the growing literature refuting of the long held belief that spaces such as the CNS are sterile. The authors should specifically address this issue up front.

The authors could more clearly discuss the opportunistic potential for the pathogen- that the fungus may be more prevalent once there is damage. The way the information is presented, it often seems that the fungi are the instigator rather than an opportunist that expands in the setting of disrupted tissues.  If truly present, is the fungus the inciting force for disease, an incidental, or functioning in an additive/synergistic manner that leads to disease progression.

There are some typos, such as "organs" (instead of 'organ') line 558

Line 255 "The text continues here" is an error.

Author Response

    • Reviewer 3

    The review contributes to the growing literature refuting of the long held belief that spaces such as the CNS are sterile. The authors should specifically address this issue up front.
  • R: We agree with the reviewer, but addressing the issue of the belief in CNS sterility was not our priority because there many studies have already discussed microbial infections, such as viruses and bacteria (ref. 85 and 87), of the CNS. Here, we wanted to point out that it was unexpected to find Malassezia in the CNS.
  • The authors could more clearly discuss the opportunistic potential for the pathogen- that the fungus may be more prevalent once there is damage. The way the information is presented, it often seems that the fungi are the instigator rather than an opportunist that expands in the setting of disrupted tissues. If truly present, is the fungus the inciting force for disease, an incidental, or functioning in an additive/synergistic manner that leads to disease progression.
  • R: Current knowledge is extremely limited on the opportunistic or pathogenic potential role of Malassezia on the CNS. In the discussion, we highlighted the importance of future studies to fully understand the role of Malassezia in neurodegenerative diseases. Here, we have tried to discuss the data on the detection of Malassezia on the CNS. It is important to know that not only Malassezia is detected but also other fungi are detected along with Malassezia. Therefore, it is difficult to dissect the role of each particular fungus. We think that if it is really present, it probably plays an additive/synergistic role with other fungi in the disease progression.
  • In the discussion section, we have added the sentences: ‘. If truly present, is the fungus the inciting force for disease, an incidental, or functioning in an additive/synergistic manner that leads to disease progression? Are Malassezia the instigator of the disease or an opportunist that colonizes tissues that have been injured by another factor?’
  • There are some typos, such as "organs" (instead of 'organ') line 558
  • R: Corrected.
  • Line 255 "The text continues here" is an error.
  • R: Corrected.